# A Pipeline for Interpretable Clinical Subtyping with Deep Metric Learning

**Haoran Zhang** [* 1]   **Qixuan Jin** [* 1]   **Thomas Hartvigsen** [* 2]   **Miriam Udler** [3]   **Marzyeh Ghassemi** [1]

## Abstract

Clinical subtyping, a critical component of personalized medicine, classifies patients with a particular disease into distinct subgroups based on their unique features. However, conventional data-driven subtyping approaches often entail a manual characterization of the identified clusters, complicating the task due to the high dimensionality and heterogeneity of the data. In this work, we propose a novel framework for interpretable clinical subtyping using deep metric learning. Our proposed pipeline unifies prior approaches to clinical subtyping, and introduces automatic characterization of the learned clusters in an interpretable and clinically meaningful manner. We showcase the effectiveness of this framework on real-world clinical case studies, demonstrating its utility in uncovering actionable clinical knowledge.

## 1. Introduction

As machine learning models reach or surpass human level performance on many clinical predictive tasks (Liu et al., 2019), they are being increasingly deployed in real-world clinical settings (Sendak et al., 2020). However, in addition to simple prediction in supervised tasks, machine learning also has the promise to uncover and extract useful clinical knowledge and clinical *insights*. Indeed, prior works have used machine learning to tailor personalized treatment plans (Coronato et al., 2020), learn complex physiological relationships (Qian et al., 2021), and discover causal mechanisms (Seedat et al., 2022; Hasan & Gani, 2022).

In this work, we focus on the problem of clinical *subtyping*, also known as clinical phenotyping (Yang et al., 2023). In clinical subtyping, given a labelled dataset containing patients who do and do not have some disease of interest,

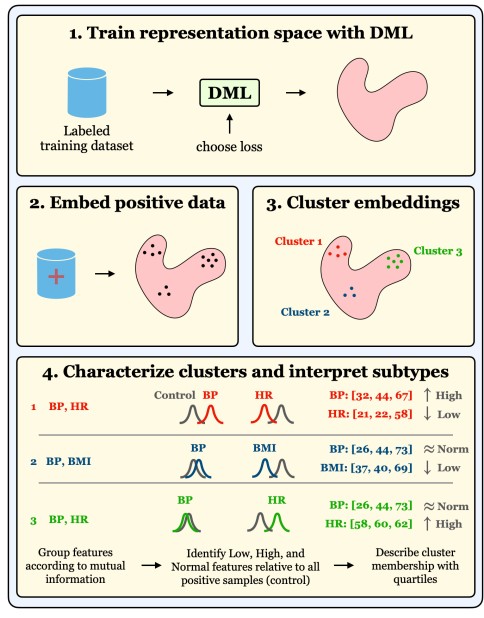

*Figure 1.* Pipeline for subtyping clinical conditions with DML.

the goal is to categorize those who have the disease into distinct subgroups, based on their input features. These subtypes can then be used to inform future treatments plans, predict disease trajectories and risks of complications, and ultimately gain a greater understanding of how the disease manifests in different individuals (Banda et al., 2018).

Clinical subtypes have long been used in medicine, from asthma (Bel, 2004), to acute kidney injury (Makris & Spanou, 2016), to COPD (Miravitlles et al., 2013). However, the majority of existing clinical subtypes have been manually derived by panels of experts using clinical expertise, which is time-consuming and also does not yield a quantifiable evaluation metric. Recently, many works have derived clinical subtypes using data-driven approaches, e.g. for sepsis (Knaus & Marks, 2019), diabetes (Banda et al., 2017), hypertension (Sweatt et al., 2019), mental illnesses (Benoit et al., 2020), and heart failure (Urban et al., 2022). These approaches often involve clustering the representations learned by a supervised machine learning model (See Steps 1-3 in Figure 1). However, these works still require a

---

[*]Equal contribution   [1]MIT   [2]University of Virginia   [3]Massachusetts General Hospital. Correspondence to: Haoran Zhang <haoranz@mit.edu>.

*Workshop on Interpretable ML in Healthcare at International Conference on Machine Learning (ICML)*, Honolulu, Hawaii, USA. 2023. Copyright 2023 by the author(s).

manual investigation to *characterize* the behavior of these clusters. This characterization process can poses significant challenges due to the high dimensionality and heterogeneity of the data.

In this work, we propose a pipeline for interpretable clinical subtyping using Deep Metric Learning (DML) (Kaya & Bilge, 2019). Specifically,

- We unify approaches from prior works to propose a pipeline for clinical subtyping. Crucially, unlike prior work, our pipeline automatically characterizes learned clusters to be interpretable and clinically meaningful.

- We demonstrate our pipeline on two real-world clinical case studies: Type 2 diabetes subtyping using the All of Us dataset (Ramirez et al., 2022), and Shock subtyping using the MIMIC-III dataset (Johnson et al., 2016). We find that our derived subtypes uncover clinical insights found in previous medical studies, and may be useful in informing downstream clinical action.

## 2. Related Work

### 2.1. Deep Metric Learning

Deep Metric Learning (DML) is a leading approach for modeling the similarity between data. Some of the most notable successes have been zero-shot retrieval (Oh Song et al., 2016a; Wu et al., 2017; Roth et al., 2020), clustering (Ge, 2018; Sohn et al., 2019), verification (Deng et al., 2019; Liu et al., 2017), and few-shot (Snell et al., 2017), contrastive (Khosla et al., 2020), and unsupervised representation learning (He et al., 2020; Chen et al., 2020). DML methods have also been proposed for broad-randing machine learning tasks including multitask learning (Opitz et al., 2017; 2018; Milbich et al., 2020; Milbich et al., 2020; Xuan et al., 2018; Kim et al., 2018), multi-modality (Roth et al., 2022), feature mining (Roth et al., 2019; Sanakoyeu et al., 2019), and adversarial regularization (Lin et al., 2018; Zheng et al., 2019; Duan et al., 2018; Ko et al., 2021; Sinha et al., 2020; Milbich et al., 2021). Our pipeline is compatible with all recent DML approaches, though methods and tasks should be paired appropriately when possible.

### 2.2. Interpretable Clinical Subtyping

Machine learning is a promising direction for subtyping clinical conditions (Baytas et al., 2017; Zhang et al., 2019; Brendel et al., 2021; Castaldi et al., 2020; Banerjee et al., 2021; Cascianelli et al., 2020; An et al., 2022; Su et al., 2021; Lu et al., 2018; Sinkala et al., 2020). For example, subtypes have been identified for Parkinson's Disease (Brendel et al., 2021; Faghri et al., 2018), Heart Failure (Banerjee et al., 2021), Alzheimer's Disease (An et al., 2022), Schizophrenia (Chand et al., 2020), and even SARS-CoV-2 (Zhang et al., 2023). Across the board, there are a wide range of techniques used to learn these subtypes and there is a long history of success built on traditional clustering methods (Ieva et al., 2017; Ather et al., 2011; Kao et al., 2015; Ahmad et al., 2014; Panahiazar et al., 2015; Vellone et al., 2017). Since DML generalizes classic clustering approaches, recent methods have also begun adopting DML (Liu et al., 2021; 2023; Qi et al., 2021; Tian et al., 2019). However, successfully using DML for sub-typing is harder than traditional clustering, since DML clusters are in latent spaces. To the best of our knowledge, no works have studied the need to re-integrate interpretability into the clinical subtyping pipeline when clustering via DML.

## 3. Problem Setup

We frame the clinical subtyping problem as follows. We are given as input a dataset $X \in \mathbb{R}^{n \times d}$, with binary disease labels $Y \in \{0, 1\}^n$. The goal is to assign each patient with a positive label ($\{i : Y_i = 1\}$) to a subtype $C = \{1, ..., c\}^n$, and then to *characterize* these subtypes. One challenge of automatically coming up with a characterization is the high dimensionality of the feature space (i.e. $d$ may be large). To alleviate this problem, we make use of *feature groups*, which are natural (or domain-defined) groupings of features. For example, the feature group *blood pressure* may contain features such as {*mean systolic BP 1 year prior, max diastolic BP 6 months prior, ...*}. In the simplest case, each feature group contains exactly one feature. Given these feature groupings, a characterization can then be created using the values of features in each feature grouping for each subtype. We present one form of such a characterization in Section 4. One important consideration is that the feature values of a subtype should be compared both against other subtypes, but also against the controls (i.e. patients with a negative label). For example, for a particular feature group, a cluster may have "high" feature values relative to other subtypes, but "normal" feature values relative to those without the condition.

## 4. Proposed Pipeline

Below, and in Figure 1, we describe our proposed pipeline for clinical phenotyping using deep metric learning.

**Step 1: Learn a DML Model.** First, we learn a DML model. We choose DML in particular over conventional supervised learning, as DML models have been shown to learn a better-structured representation space (Roth et al., 2020) than Empirical Risk Minimization (ERM).

**Step 2: Embed Positive Test-Set Samples.** We compute embeddings for test-set patients with the condition.

**Step 3: Cluster Embeddings.** We use a clustering algorithm to label each positive patient with a subtype $\{1, ..., c\}$. Here, $c$ is a hyperparameter that can be chosen with domain knowledge, cross validation (Palacio-Niño & Berzal, 2019), or extrinsic metrics (Amigó et al., 2009).

**Step 4: Characterize Clusters.** We characterize the patients in the learned clusters, by showing a table of dimension $k \times c$, where $k$ is a hyperparameter. Each cell of the table contains a comparison of the feature values of the cluster for the particular feature group, relative to both the controls and other clusters.

**Step 4.1: Feature Group Selection.** We start by selecting the $k$ feature groups that are most informative in deciding the subtype. We use a mutual information based selection criteria (Estévez et al., 2009; Vergara & Estévez, 2014). In particular, given a feature group $\phi$, we compute $\max_{i \in \phi} I(X[:, i]; C)$. Then, we select the $k$ feature groups with the largest value.

**Step 4.2: Comparison to Controls.** Here, we wish to quantify whether the feature values of a particular cluster are "normal" relative to the controls. Note that we do not wish to measure whether the two sets of samples are drawn from the same distribution, as a cluster feature distribution that concentrates heavily on the mean of the control distribution is intuitively normal, but would be flagged by this test. Instead, we use the overlapping index (Pastore & Calcagnì, 2019; Yitzhaki, 1994), which measures the area of overlap between two empirical distributions. We state that the feature group value is normal (denoted with $\approx$) for a particular cluster if its average overlapping index is greater than 0.5, and is abnormal otherwise. For abnormal feature groups, we evaluate whether the mean of the feature values is higher than the controls, and report whether it is abnormal high ($\uparrow$) or abnormal low ($\downarrow$).

**Step 4.3: Comparison to Other Clusters.** Finally, we compare the feature values of each cluster against other clusters. For feature groups containing binary features, we report the average value of each cluster across features. For continuous feature groups, we report the quartiles of the percentile values averaged across feature values relative to the marginal distribution containing all positive patients.

## 5. Case Studies

We demonstrate the utility of our pipeline in two real-world clinical case studies. Here, we present a case study for phenotyping type 2 diabetes 2-years prior to diagnosis using longitudinal EHR data. In Appendix A, we present another case study for phenotyping Shock in the ICU.

*Table 1.* Evaluation of embedding quality and downstream prediction accuracy of ERM versus various DML models.

| | Recall@1 | Recall@2 | NMI | F1 | Lin AUROC |
|---|---|---|---|---|---|
| **ERM** | 0.774 | 0.881 | 0.247 | 0.642 | 0.899 |
| **Triplet** | 0.768 | 0.875 | 0.351 | 0.723 | 0.901 |
| **N-Pair** | **0.779** | **0.883** | **0.368** | **0.728** | **0.909** |
| **Lifted** | 0.765 | 0.881 | 0.333 | 0.712 | 0.907 |
| **ProxyNCA** | 0.758 | 0.875 | 0.339 | 0.712 | 0.880 |

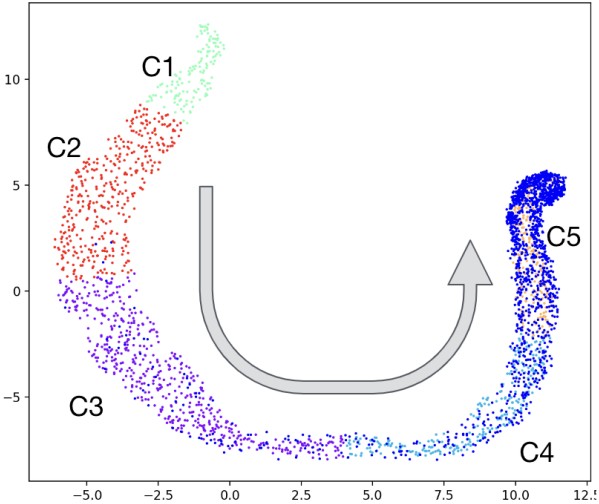

*Figure 2.* UMAP projection of test-set samples. Points in dark blue (·) represent control group samples. Remaining points represent diabetic patients from each learned subtype.

### 5.1. Type 2 Diabetes Subtyping

**Data** We use the All of Us dataset (Ramirez et al., 2022), which contains longitudal EHR records for over 500,000 patients from across the US. We construct a cohort of patients with Type 2 Diabetes (T2D), as well as controls, using the eMERGE algorithm (Wei et al., 2012). As features, we use chemical measurements, physical measurements, conditions, prescriptions, and demographics (see Table 4 for a full list). Unlike prior work (Li et al., 2015), we seek patients who *will* develop T2D in two years, so we set the censor date to be two years prior to the diagnosis date for diabetic patients. We flatten the time-series by computing the mean, max, and min values of each measurement for the 6 months, 2 years, and entire history prior to the censor date. We then train a linear model to predict case from control, and select control patients with the highest predicted risk of T2D to ensure a hard-enough learning task to produce a high-quality representation space. Thus our dataset has 15,134 patients, 50% of whom will develop T2D in 2 years.

**DML Models** We train a variety of DML models, varying the loss type (triplet (Hoffer & Ailon, 2015), N-pair (Sohn, 2016), Lifted (Oh Song et al., 2016b), Prox-

*Table 2.* Characterization of subtypes learned. The following notation compares feature values to the control set: ↑(abnormal high), ↓(abnormal low), ≈(same as control). Numbers in square brackets denote either quartiles of percentiles relative to the marginal, or the average incidence rate. (%) denotes the percentage of positive test samples belonging to that cluster.

| | C1 (9.1%) | C2 (26.0%) | C3 (34.8%) | C4 (16.3%) | C5 (13.9%) |
|---|---|---|---|---|---|
| **Blood pressure** | ↑[31.63, 44.09, 67.33] | ↑[32.31, 44.49, 64.75] | ≈[26.23, 44.36, 73.16] | ≈[12.20, 44.56, 82.26] | ≈[12.60, 75.77, 89.04] |
| **BMI** | ↓[37.48, 37.48, 69.41] | ↓[37.62, 37.62, 37.62] | ≈[37.48, 37.48, 70.99] | ≈[37.55, 37.55, 86.13] | ≈[37.55, 72.73, 89.64] |
| **Heart rate** | ↓[21.56, 21.56, 58.31] | ↓[21.63, 58.24, 58.24] | ↑[58.44, 58.44, 58.44] | ≈[21.43, 58.24, 84.73] | ≈[21.03, 57.78, 91.00] |
| **GERD** | ↑0.54 | ↑0.28 | ↑0.12 | ↑0.08 | ↓0.08 |
| **Neutrophils** | ↑[22.31, 51.17, 74.08] | ↑[24.83, 54.54, 77.10] | ≈[25.63, 50.47, 73.77] | ↓[24.27, 43.16, 73.63] | ≈[27.42, 48.46, 77.33] |
| **Asthma** | ↑0.40 | ↑0.16 | ↑0.07 | ↓0.05 | ↓0.06 |
| **Body weight** | ≈[38.95, 38.95, 76.89] | ≈[38.89, 38.89, 71.36] | ≈[38.89, 38.89, 70.29] | ≈[38.89, 38.89, 76.65] | ≈[11.92, 66.69, 81.61] |
| **Hypertension** | ↑0.39 | ↑0.21 | ↑0.11 | ↑0.06 | ↓0.05 |
| **Creatinine** | ≈[30.51, 57.44, 61.42] | ≈[31.78, 57.51, 79.24] | ≈[25.97, 57.31, 74.83] | ≈[21.97, 57.18, 57.18] | ≈[21.31, 57.31, 75.15] |
| **Smoking** | ↑0.22 | ↑0.16 | ↑0.08 | ↓0.05 | ↓0.06 |

yNCA (Kim et al., 2020)), the latent dimension, and other optimization and model architecture hyperparameters. We select the model with the highest validation downstream AUROC. In Table 1, we find that DML methods outperform ERM, both in representation quality, and downstream classification. We select the best performing N-Pair model for further analyses.

**Phenotypes** We utilize our proposed pipeline to derive phenotypes, using K-means with $c = 5$ (informed by prior work (Landi et al., 2020)), and $k = 10$. In Figure 2, we learn a UMAP embedding (McInnes et al., 2018) on the diabetic patients, and project all samples from the test-set. In Table 2, we show the subtype characterization from our pipeline. First, we find that Clusters 1 and 2 consist of patients that one would typically consider to be pre-diabetic – high blood pressure, with cardiovascular diseases (Einarson et al., 2018), GERD (Sun et al., 2015), and asthma (Torres et al., 2021), all of which have been associated with T2D in prior work.

Next, we find that Clusters 4 and 5 consists of patients that appear to be most healthy, and control groups also embed most similarly to these clusters. These patients also exhibit greater heterogeneity in blood pressure, BMI, and heart rate. Effective identification of patients belonging to this cluster may result in taregeted intereventions to prevent T2D onset (Satterfield et al., 2003; Kriska, 2003).

Finally, we find that the clusters we have characterized exhibit some similarities to those presented in prior work. For example, our Cluster 3 most similarly resembles Subgroup 1 identified in Landi et al. (2020), and our Cluster 1 most similarly resembles Subgroup 3 identified in Li et al. (2015). However, we emphasize that our pipeline allows for the characterization of subtypes to be automated.

**Downstream Complications** One utility of clinical subtypes is to stratify risk for downstream complications. In Figure 3, we plot the incidence rate of various complica-

tions, *after* each patient's T2D diagnosis. We find that, as expected, Cluster 1 has the highest risk of all complications relative to other clusters, especially disorders of the nervous system. We also find that Cluster 5, though relatively healthy, has a relatively high risk of chronic kidney disease. Knowledge of these potential complications can yield targeted treatments and interventions.

## 6. Discussion

Our results confirm that DML is a powerful approach to clinical subtyping. Leveraging the flexibility of DML representations enables more realistic subtypes by allowing us to learn from more types of data. For example, future works may improve multi-modal or temporal subtyping, for which traditional clustering methods are ill-suited. The lack of inherent interpretability in most DML setups, however, tempers their use. Our work pushes in this direction by clearly highlighting the need to characterize DML-identified clusters, and we showcase our approach to solving this problem. By comparing feature statistics across DML-clustered data, we indeed find that our approach can discover and interpret subtypes in Type 2 Diabetes. We are currently collaborating with clinicians to verify these subtypes.

**Future Work** There are several areas of future work. First, in the T2D case study, we found that our original control group was too "easy", and increased the task difficulty by subsetting to hard control samples. Better understanding of how negative samples influence learned representations and subtypes would allow us to integrate control group customization as a "Step 0" of our pipeline, and this has been an area of study in core contrastive learning literature (Zhang & Stratos, 2021; Kalantidis et al., 2020). Second, we would like to evaluate the quality of our subtypes using extrinsic metrics. For example, intuitively, subtypes are more useful if they exhibit different downstream complications and reactions to treatments. These metrics could then inform hyperparameter selection in Steps 1-3.

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

## A. Case Study: Shock Phenotyping in MIMIC-III

**Data** We use the MIMIC-III database (Johnson et al., 2016), which consists of ICU records for adult patients admitted to critical care units of the Beth Israel Deaconess Medical Center in Boston between 2001 and 2012. Specifically, we use the phenotyping cohort created by Harutyunyan et al. (2019), and choose Shock as the illness of interest. The cohort consists of 30,558 patients, 2,204 of which had shock. We bin the labs and vitals provided in the dataset into 1 hour bins (up to 48 hours), and train DML models using GRU (Dey & Salem, 2017) as the backbone.

**Phenotypes** We run our pipeline using $c = 3$ and $k = 8$. We present the learned phenotypes in Table 3.

*Table 3.* Characterization of subtypes learned. The following notation compares feature values to the control set: ↑(abnormal high), ↓(abnormal low), ≈(same as control). Numbers in square brackets denote either quartiles of percentiles relative to the marginal. (%) denotes the percentage of positive test samples belonging to that cluster.

|  | **C1** (44.1%) | **C2** (28.4%) | **C3** (27.4%) |
|---|---|---|---|
| **Oxygen saturation** | ↓[13.98, 32.23, 54.50] | ↓[24.64, 52.61, 74.88] | ↓[36.61, 67.06, 86.97] |
| **Mean blood pressure** | ↓[16.82, 34.12, 59.48] | ≈[25.59, 45.50, 67.77] | ≈[45.85, 70.38, 88.27] |
| **Glucose** | ↑[45.26, 63.27, 79.38] | ≈[22.27, 44.08, 72.99] | ≈[22.39, 40.88, 62.68] |
| **Respiratory rate** | ↑[33.53, 63.27, 82.82] | ≈[23.22, 44.08, 68.25] | ≈[25.00, 50.71, 75.95] |
| **Systolic blood pressure** | ↓[20.14, 38.15, 55.92] | ↓[21.33, 49.76, 72.04] | ≈[39.10, 68.96, 86.85] |
| **Diastolic blood pressure** | ≈[21.45, 38.15, 59.24] | ≈[21.80, 47.87, 70.62] | ≈[49.05, 69.43, 86.14] |
| **Heart rate** | ≈[30.09, 48.58, 77.73] | ≈[20.85, 45.97, 72.04] | ≈[32.46, 53.32, 75.95] |
| **Glasgow coma scale total** | ↓[21.09, 52.13, 80.09] | ↓[24.64, 52.61, 80.09] | ↓[31.87, 49.53, 80.09] |

## B. Additional Results: All of Us

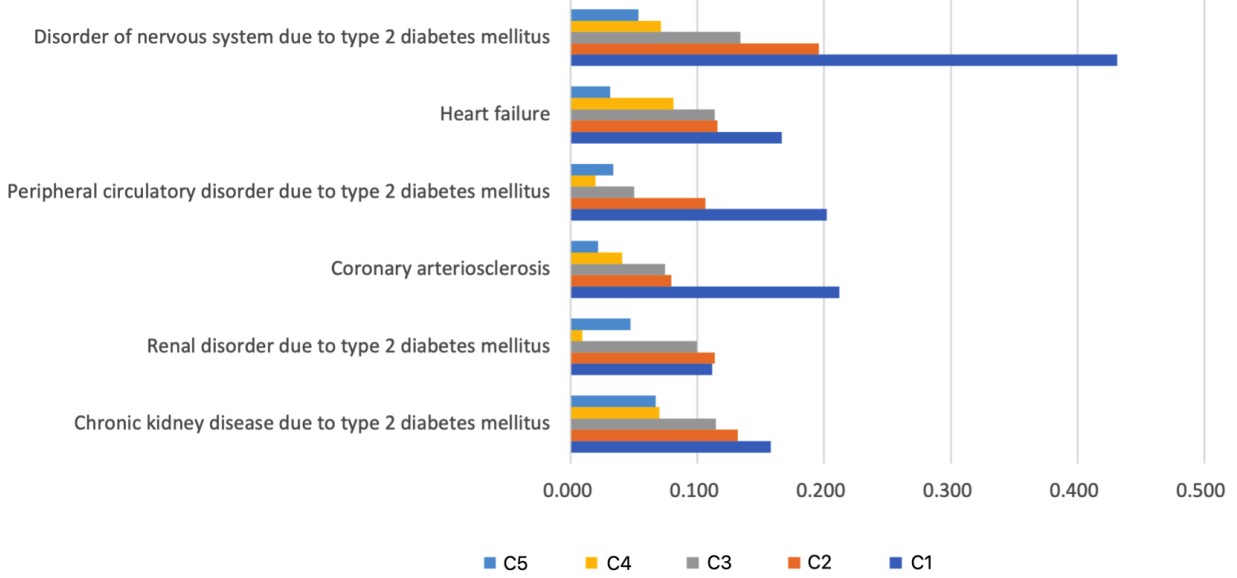

*Figure 3.* Incidence rate of various complications for each cluster *after* each patient's T2D diagnosis date. We find that C1 (previously identified as the most serious cluster) is at highest risk of complications, especially disorders of the nervous system.

*Table 4.* List of feature used in the All of Us model.

| Feature Source | Feature Groups |
| --- | --- |
| Conditions | Acquired hypothyroidism, Actinic keratosis, Acute pharyngitis, Acute upper respiratory infection, Alcohol dependence, Allergic rhinitis, Allergic rhinitis due to pollen, Anemia, Anxiety disorder, Asthma, Atherosclerosis of coronary artery without angina pectoris, Atrial fibrillation, Benign essential hypertension, Bipolar disorder, Cardiac arrhythmia, Carpal tunnel syndrome, Chronic kidney disease stage 3, Chronic obstructive lung disease, Complication due to diabetes mellitus, Congestive heart failure, Coronary atherosclerosis, Depressive disorder, Diabetes mellitus without complication, Disorder of bone, Disorder of muscle, Dysphagia, End-stage renal disease, Eruption, Essential hypertension, Gastroesophageal reflux disease, Gastroesophageal reflux disease without esophagitis, Generalized anxiety disorder, Human immunodeficiency virus infection, Hyperglycemia due to type 2 diabetes mellitus, Hyperlipidemia, Hypothyroidism, Inflammatory dermatosis, Insomnia, Iron deficiency anemia, Kidney stone, Lumbago with sciatica, Major depression, single episode, Migraine, Mixed hyperlipidemia, Moderate recurrent major depression, Morbid obesity, Multiple sclerosis, Myopia, Nausea and vomiting, Nicotine dependence, Nuclear senile cataract, Obesity, Obstructive sleep apnea syndrome, Osteoarthritis, Osteoarthritis of knee, Osteoporosis, Paroxysmal atrial fibrillation, Posttraumatic stress disorder, Presbyopia, Primary malignant neoplasm of female breast, Primary malignant neoplasm of prostate, Pure hypercholesterolemia, Recurrent major depression, Rheumatoid arthritis, Senile hyperkeratosis, Sleep apnea, Spinal stenosis of lumbar region, Systemic lupus erythematosus, Tobacco dependence syndrome, Type 2 diabetes mellitus, Type 2 diabetes mellitus without complication, Uncomplicated asthma, Urinary tract infectious disease, Vitamin B deficiency, Vitamin D deficiency |
| Physical Measurements | Body height, Body mass index (BMI) [Ratio], Body weight, Diastolic blood pressure, Heart rate, Systolic blood pressure |
| Chemical Measurements | Alanine aminotransferase [Enzymatic activity/volume] in Serum or Plasma, Aspartate aminotransferase [Enzymatic activity/volume] in Serum or Plasma, Calcium [Mass/volume] in Serum or Plasma, Carbon dioxide, total [Moles/volume] in Serum or Plasma, Chloride [Moles/volume] in Serum or Plasma, Creatinine [Mass/volume] in Serum or Plasma, Erythrocyte distribution width [Ratio] by Automated count, Glucose [Mass/volume] in Serum or Plasma, Hemoglobin [Mass/volume] in Blood, Leukocytes [#/volume] in Blood by Automated count, MCH [Entitic mass] by Automated count, MCHC [Mass/volume] by Automated count, Neutrophils [#/volume] in Blood by Automated count, Sodium [Moles/volume] in Serum or Plasma, Urea nitrogen [Mass/volume] in Serum or Plasma |
| Drugs | 1 ML hydromorphone hydrochloride 1 MG/ML Injection, 10 ML sodium chloride 9 MG/ML Prefilled Syringe, 1000 ML sodium chloride 9 MG/ML Injection, 2 ML fentanyl 0.05 MG/ML Injection, 2 ML ondansetron 2 MG/ML Injection, NDA020503 200 ACTUAT albuterol 0.09 MG/ACTUAT Metered Dose Inhaler, NDA021457 200 ACTUAT albuterol 0.09 MG/ACTUAT Metered Dose Inhaler, acetaminophen, acetaminophen 325 MG / hydrocodone bitartrate 5 MG Oral Tablet, acetaminophen 325 MG / oxycodone hydrochloride 5 MG Oral Tablet, acetaminophen 325 MG Oral Tablet, acetaminophen 500 MG Oral Tablet, albuterol, amlodipine, amlodipine 10 MG Oral Tablet, amlodipine 5 MG Oral Tablet, amoxicillin, ascorbic acid, aspirin, aspirin 81 MG Delayed Release Oral Tablet, atorvastatin, atorvastatin 20 MG Oral Tablet, atorvastatin 40 MG Oral Tablet, bupropion, calcium carbonate, calcium chloride 0.0014 MEQ/ML / potassium chloride 0.004 MEQ/ML / sodium chloride 0.103 MEQ/ML / sodium lactate 0.028 MEQ/ML Injectable Solution, cholecalciferol, cyclobenzaprine, cyclobenzaprine hydrochloride 10 MG Oral Tablet, diphenhydramine hydrochloride 50 MG/ML Injectable Solution, docusate, docusate sodium 100 MG Oral Capsule, fentanyl, fentanyl 0.05 MG/ML Injection, fluticasone, fluticasone propionate 0.05 MG/ACTUAT Metered Dose Nasal Spray, furosemide, gabapentin, gabapentin 300 MG Oral Capsule, heparin, heparin sodium, porcine 5000 UNT/ML Injectable Solution, hydrochlorothiazide, hydrochlorothiazide 25 MG Oral Tablet, hydromorphone, ibuprofen, ibuprofen 600 MG Oral Tablet, levothyroxine, lidocaine, lidocaine hydrochloride 10 MG/ML Injectable Solution, lisinopril, lisinopril 10 MG Oral Tablet, loratadine 10 MG Oral Tablet, lorazepam, metoprolol, midazolam, midazolam 1 MG/ML Injectable Solution, omeprazole, omeprazole 20 MG Delayed Release Oral Capsule, ondansetron, ondansetron 2 MG/ML Injectable Solution, oxycodone, oxycodone hydrochloride 5 MG Oral Tablet, pantoprazole, pantoprazole 40 MG Delayed Release Oral Tablet, polyethylene glycol 3350 17000 MG Powder for Oral Solution, potassium chloride, prednisone, ranitidine, sertraline, simvastatin, sodium chloride, sodium chloride 9 MG/ML Injectable Solution, sodium chloride 9 MG/ML Injection, tamsulosin hydrochloride 0.4 MG Oral Capsule, tramadol hydrochloride 50 MG Oral Tablet |
| Demographics | Age, Gender |

