# OpenReview forum: "A Pipeline for Interpretable Clinical Subtyping with Deep Metric Learning"
_ICML.cc/2023/Workshop/IMLH — IMLH 2023 OralShortPaper_

### Official Review · Reviewer_j8HA · 2023-06-10
**Overall, the work is quite original and significant due to its innovative application of DML to an important healthcare problem. The results and performance, backed by real-world case studies, are promising and could have implications for personalized healthcare. However, the complexity of the pipeline and the assumptions made about data quality and feature grouping may limit its applicability in certain scenarios.**

**Rating:** 8
**Confidence:** 3

**Review:**

The work focuses on an important area in the field of healthcare - clinical subtyping or clinical phenotyping - using a combination of Deep Metric Learning (DML) and clustering techniques. It offers a pipeline that not only categorizes patients into subgroups based on their disease manifestations but also characterizes these clusters to provide meaningful insights for medical practitioners. The work has successfully applied the pipeline on real-world clinical datasets and shown the derived subtypes are in line with the results of previous medical studies.

Pros:

Importance of Topic: Clinical subtyping is a critical area in medical research as it provides insight into how diseases manifest in different patients, allowing for personalized treatment plans. This paper addresses this issue in a systematic way.

Innovative Approach: The use of DML is innovative as it has been proven to work well in modeling similarity between data, and this application in clinical subtyping seems to be a fitting use of the technology.

Interpretability: The approach maintains the interpretability of the derived clinical subtypes, which is essential for their use in real-world clinical settings.

Cons:

Complexity: The methodology seems to involve quite a few steps, and each requires careful execution and parameter tuning. This could make the pipeline quite complex to implement in a practical setting.

Dependence on Feature Grouping: The approach assumes a high quality, domain-specific grouping of input features. In real-world scenarios, these groupings may not always be readily available or easily definable, limiting the applicability of the pipeline.

Choice of Clustering Algorithm: The work does not delve into the choice of the clustering algorithm used in Step 3 of the pipeline, which could impact the final results.

---

### Official Review · Reviewer_AzSS · 2023-06-12
**A Pipeline for Interpretable Clinical Subtyping with Deep Metric Learning**

**Rating:** 9
**Confidence:** 4

**Review:**

In this paper, the authors propose a novel framework for interpretable clinical subtyping using deep metric learning (DML). The proposed method is applied to two real-world clinical case studies to demonstrate its utility in uncovering actionable clinical knowledge. The motivation of the paper is clear. This work studies the need to re-integrate interpretability into the clinical subtyping pipeline when clustering via DML. However, only one baseline (i.e., ERM) is used to compare with the proposed method. More state-of-the-art methods are desired for comparison. The authors should also add several sentences to describe the baseline ERM.

---

### Meta-Review · Area_Chair_M9dA · 2023-06-18

**Recommendation:** Accept (Oral)
**Confidence:** 4

**Metareview:**

This paper proposed a novel framework for interpretable clinical subtyping using deep metric learning. All reviewers agreed that the paper is well written and motivated. The methods were sound and effective, and the evaluation is thorough. There were a few concerns raised by one reviewer regarding complexity, which the authors should take into account when preparing for the camera ready version.

Overall this paper is a solid contribution to the workshop. I recommend acceptance.

---

### Decision · Program_Chairs · 2023-06-20

Accept (Oral Short Paper)